# On the Advantages of Microwave Photonic Interrogation of Fiber-Based Sensors: A Noise Analysis

**DOI:** 10.3390/s23073746

**Published:** 2023-04-04

**Authors:** Ulrich Nordmeyer, Torsten Thiel, Konstantin Kojucharow, Niels Neumann

**Affiliations:** 1Institute of Electrical Information Technology, Clausthal University of Technology, 38678 Clausthal-Zellerfeld, Germany; 2Advanced Optics Solutions (AOS) GmbH, Overbeckstr. 39a, 01139 Dresden, Germany; 3Kojucharow Microwave Development and Components (KMDC), Zur Bleiche 15, 01279 Dresden, Germany

**Keywords:** electrical readout, fiber sensor, fiber bragg grating, microwave photonics, phase response, radio-over-fiber, SNR, wireless sensing

## Abstract

Although microwave photonic approaches have been used for fiber sensing applications before, most contributions in the past dealt with evaluating the sensor signal’s amplitude. Carrying this topic on, the authors previously presented a scheme for the interrogation of fiber sensors that was based on a fiber Bragg grating’s phase response for the electrical signal. However, neither has the measurement setup been analyzed nor have the amplitude and phase-based approaches been compared in detail before. Hence, this paper picks up the previously proposed setup, which relies on an amplitude modulation of the optical signal and investigates for sources of signal degradation, an aspect that has not been considered before. Following the incorporation of the microwave signal, the setup is suitable not only for an amplitude-based evaluation of fiber Bragg gratings but also for a phase-based evaluation. In this context, the signal-to-noise ratios are studied for the conventional amplitude-based evaluation approach and for the recently developed phase-based approach. The findings indicate a strong advantage for the signal-to-noise ratio of the phase response evaluation; an 11 dB improvement at the least has been found for the examined setup. Further studies may investigate the consequences and additional benefits of this approach for radio-over-fiber sensing systems or general performance aspects such as achievable sensitivity and sampling rates.

## 1. Introduction

The field of microwave photonics (MWP) has been subject to research interests for many years. Among a very large overall count of publications associated with MWP, some are dedicated to presenting an overview of this research field [1,2,3,4,5]. Reviewing the latest advances at the time of their writing, they cover different application areas such as communication technology, signal generation, signal processing and integrated photonics. At the same time, the sub-field of sensors and measurement tasks is still underrepresented. However, the publication history of the past decade shows increasing research work in this area. A partial overview is given by Yao [6], while other studies are linked to a variety of specific problem formulations such as distributed fast fiber-optic sensing [7], multiplexing sensors [8] and measuring properties of microwave signals [9]. Further work concentrates on the microwave-based interrogation of fiber grating sensors, e.g., aiming for resolution and speed improvements [10], high-resolution multi-point sensing [11], antigen biosensing [12] and wireless sensing schemes [13], all of which are incorporating Fiber Bragg Gratings (FBGs) as sensing elements. Earlier, more general studies on the microwave interrogation of fiber-optic devices began investigating the use of MWP for optical network analysis. The scheme was based on single-sideband (SSB) modulation [14,15] and afterwards adapted for measuring and reconstructing the phase response of FBGs [16,17]. The same basic principle was utilized for an improved evaluation of quasi-distributed sensing elements [18,19]. A similar concept involves double-sideband (DSB) modulation, and its applicability has been reviewed along with system modeling considerations and functionality examples [20].

In addition to these publications, the authors of this paper have also contributed to the topic of MWP-based FBG characterization and fiber-optic sensing in their previous studies. Starting with a presentation of the general wireless evaluation approach for FBGs [21] and a demonstration of pH sensing in such a radio-over-fiber (RoF) setup [22], the work was continued by researching a novel scheme of electrical read-out in the form of utilizing the group-delay characteristic as an alternative sensor characteristic to the conventionally used amplitude response [23]. The applicability of this evaluation scheme to thermometry has been evaluated [24] and the effects resulting from the DSB modulation have been studied and modeled in detail [25].

In contrast to the conventional approach, the authors’ motivation was to make use of the electrical phase information that is present in MWP sensing setups but remained unused before. That means, other aspects that have been investigated before, such as improving vector analyzing of optical networks, precisely reconstructing the actual dispersion characteristic of FBGs or dealing with multi-point or quasi-distributed sensing of a large number of fiber sensors, are not in focus. An optimization of the novel approach led to simplified experimental setups and finally to a direct evaluation of the electrical phase, which for FBGs possesses a dependency on the measurand. After demonstrating the general suitability of this concept for sensing applications [26], the present paper is focused on analyzing the interrogation scheme for possible signal impairments and their consequences regarding measurement accuracy. The setup used previously for the general suitability study is picked up and the possible noise performance is analyzed in detail for the first time. Based on these considerations, the conventional amplitude characteristic read-out method is compared to the phase evaluation method, and both advantages and disadvantages are discussed. The study starts with the explanation of the basic concept of microwave photonic FBG interrogation and continues with the presentation of the experimental setup and the measurement results in Section 2. A detailed analysis of signal degradation sources and their consequences for the signal-to-noise ratio (SNR) of the respective signal follows in Section 3. Finally, the work is summarized, the core results are presented and starting points for further investigations are pointed out in Section 4.

## 2. Sensor Evaluation in the Electrical Domain

The evaluation of FBGs that are employed as optical sensors is conventionally carried out in the optical domain. Light, e.g., from a laser source, is fed into the sensor, and the power of the reflected or transmitted light is analyzed. This can be realized in a variety of ways. A source with a broad spectrum can be used, and the changes introduced by the FBG, which depend on the measurand and the FBG itself, can be analyzed with an optical spectrum analyzer (OSA). Alternatively, a tuneable laser source (TLS) can be used as the optical source and an optical power meter as the evaluation unit. Then, through a wavelength sweep of the TLS, the spectrum of the amplitude characteristic can be monitored. Both methods are costly in terms of hardware expenses and sampling time. A simpler but in general less accurate method employs a fixed wavelength laser source and a power meter. Provided that an initial state of the sensor is known or that the sensor characteristic is fully unambiguous, this scheme allows for the possible relative measurements to be translated into absolute ones.

In the following, the alternative idea of an MWP-supported evaluation of fiber-based sensors is presented.

### 2.1. Basic Concept

The fundamental concept of evaluating a fiber sensor in the electrical domain is based on the introduction of a microwave signal, as discussed by the authors before in detail [21,22,23,24,25,26]. This microwave signal modulates the light source, and the modulated light is fed into the sensor, which changes not only the signal’s amplitude but also its phase. Depending on the configuration, the transmitted or the reflected signal is then opto-electrically converted by a photodiode (PD). Finally, the resulting analog electrical microwave signal is digitized and processed.

### 2.2. Experimental Setup

Figure 1 shows the block diagram of the experimental setup used for the investigations in this paper. For manufacturers and models of the used components, please refer to this diagram. Further parameters and the overall signal flow are explained in the following.

A chirped FBG is used as the sensor. It is designed for a full width at half minimum (FWHM) transmission bandwidth of 8 nm around a central wavelength of 1536.5 nm. Further characteristics are its length of 7 mm, its chirp $\frac{d\Lambda }{dz}$ of 1 nm
mm^−1^ and its apodization with a Gaussian cosine function for providing a delay characteristic as linear as possible across the FWHM bandwidth. Tuning is realized by a fine-threaded adjusting screw that is coupled to a cantilever’s free tail. The FBG is bonded to the surface of the cantilever that bends convexly while tuning. The same FBG and tuning mechanism has been used in previous studies [26].

Due to the absence of a scale on the adjustment mechanism, the sensor position is monitored in the optical spectrum by a broadband optical source and an OSA via an optical circulator in the reflection configuration. An erbium-doped fiber amplifier (EDFA) is used for the broadband optical source. The OSA monitors a range from 1520 nm to 1560 nm at a resolution of 0.2 nm and 1001 points; the EDFA is set to an output power of 10.7 dBm.

A distributed feedback (DFB) laser diode (LD) provides the optical measurement signal. It is driven by a custom laser diode controller (LDC) at a forward current of 60 mA and a temperature of 21 ${}^{\circ }C$. The DFB-LD is modulated by a microwave sine signal that is generated by a signal generator. Its output is configured for a frequency of 2.45 GHz and an output power of 16 dBm. The microwave signal is split into two paths in order to calculate a phase relation, which is crucial for the evaluation scheme.

The modulated optical signal is fed via a circulator into the second port of the FBG and then opto-electrically converted. This is carried out by a custom module that consists of a biased PD and a two-stage radio frequency (RF) amplifier with a gain of 9 dB.

Both signal paths—the directly branched off reference path and the sensor path—contain anti-aliasing filters as the last element before the signals are digitized by an analog-to-digital converter (ADC) and finally processed on a personal computer (PC). The filters are necessary due to the operation of the ADC in the 17th Nyquist zone at a sampling rate of 300 MHz in single-shot mode, collecting 65,536 samples per channel for each trigger event.

Previous results of the authors [26] have compared this approach with a benchmark setup that is based on an electrical vector network analyzer (EVNA) and proven its validity. Therefore, this step does not need to be repeated here. Besides that, the focus of this paper is on the noise analysis of the ADC-based setup as some performance drawbacks come along with the advantage of lower hardware expenses compared with the EVNA-based setup.

### 2.3. Derivation of the Characteristic Curves

Furthermore, in [26], the authors have derived in detail how the characteristic curves of FBG-based sensors can be determined. To summarize, the FBG is tuned, and its spectrum is thereby swept over the fixed, small bandwidth optical signal of the modulated laser source. A width of 0.5 nm and a range from 1533 nm to 1540 nm have been chosen for the tuning parameters of the FBG’s center wavelength. After a settling time of 1 min to account for transient effects of the laser diode, five 20 s delayed single-shot measurements are triggered at each of the approached positions. The gathered data contain a time series of amplitude values for each ADC channel, i.e., for the reference signal and for the sensor signal. A sine wave is fitted to slices of both time discrete signals, resulting in parameters for frequency, amplitude, phase and offset. The fitting is carried out by an optimizer and repeated for 10 slices per single-shot measurement, with each slice having a length of 6000 samples. Finally, the mean and standard deviation for the resulting total of 50 values per sampling position (10 slices × 5 single shot measurements) are calculated for the amplitude and the phase. From the results, the amplitude characteristic can be derived directly from the sensor signal. For the phase characteristic, the sensor signal’s phase has to be related to the reference signal’s phase by subtracting the latter from the former. Finally, both characteristic curves are normalized to the center wavelength of the interval under study.

### 2.4. Experimental Outcomes

Figure 2 and Figure 3 show the measurement results for the amplitude characteristic and for the phase characteristic, respectively. The curves follow the known amplitude and phase characteristics of the sensor, and the standard deviations are small. Judging by the available resolution, the phase characteristic appears to be an injective function, which is beneficial in contrast to the clearly non-injective amplitude characteristic. Between the measured positions of the used FBG, local extrema could exist, making the phase response a non-injective function as well, which limits the achievable sensor resolution. However, the final characteristic can be specially tailored as part of the sensor design. Furthermore, the macroscopically approximate linear progression of the phase’s course provides an almost constant sensitivity for signal deviations. Possible sources for signal deviations that are inherent to the measurement setup and their impacts on the evaluation scheme are analyzed in the subsequent section. For this analysis, minimum and maximum absolute peak values of the signal magnitudes are relevant. These can be extracted from the raw data, resulting in the minimum peak voltage $V_{\mathrm{Signal},\mathrm{peak},\min}=41.50\;mV$ and the maximum peak voltage $V_{\mathrm{Signal},\mathrm{peak},\max}=72.33\;mV$.

## 3. Noise Analysis

Following the fact that utilizing the phase response of an FBG as a sensor characteristic is a novel and, up to this point, not well-researched approach, studying the advantages and disadvantages in comparison to prevalent evaluation schemes is of great interest. Therefore, sources of signal noise are identified, their impact on the measurement signal is quantified and the findings are discussed in this section.

### 3.1. Sources of Signal Degradation

The noise sources listed below refer to the setup depicted in Figure 1 and are based on the technical performance of the used devices.

#### 3.1.1. Phase Noise of the Signal Generators

The phase noise of the modulating signal generator has no effect on the amplitude evaluation provided that the amplitude characteristic is approximately linear in the range of values covered by the noise. This is because the lower sideband (LSB) and the upper sideband (USB) in this modulation regime contribute in equal shares to the sensor signal and both sidebands will jitter mirror symmetrically around the optical carrier. At the same time, phase noise of the modulation signal can influence the phase-based measurement if the reference path and the signal path have a length difference greater than the coherence length of the modulation signal. Without matching the path lengths, the coherence length of the signal has to be taken into account, which is known for electrical signal sources to be significantly longer than relevant to the setup. The ADC is sensitive to clock jitter, which translates to a sensitivity to the phase noise of the second signal generator. According to the datasheet of the ADC [27], the SNR due to clock jitter is calculated after
(1)$${\mathrm{SNR}}_{\mathrm{Jitter},\mathrm{Amplitude}}=-20\;\cdot \;\log_{10}\left(2\pi \;\cdot \;f_{\mathrm{In}}\;\cdot \;t_{\mathrm{Jitter}}\right)\;.$$

For the configured input frequency $f_{\mathrm{In}}$ of 2.45 GHz and a jitter $t_{\mathrm{Jitter}}$ according to
(2)$$t_{\mathrm{Jitter}}=\sqrt{t_{\mathrm{Jitter},\mathrm{Clock}}+t_{\mathrm{Jitter},\mathrm{ADC}\;\mathrm{Aperture}}}\;,$$
with the jitter of the clock $t_{\mathrm{Jitter},\mathrm{Clock}}$ being circa 350 fs and the internal aperture jitter of the ADC $t_{\mathrm{Jitter},\mathrm{ADC}\;\mathrm{Aperture}}$ being 100 fs, the SNR due to clock jitter ${\mathrm{SNR}}_{\mathrm{Jitter},\mathrm{Amplitude}}$ is calculated to 45.0 dB, corresponding to clock-jitter-induced noise power of $P_{\mathrm{Jitter},\mathrm{Amplitude}}=-62.7d\mathrm{Bm}$ for the weakest signal of the experiment.

#### 3.1.2. Phase Noise of the ADC

The internal aperture jitter $t_{\mathrm{Jitter},\mathrm{ADC}\;\mathrm{Aperture}}$ of the ADC generates phase noise for the quantized signal and therefore defines the SNR of the phase evaluation ${\mathrm{SNR}}_{\mathrm{Jitter},\mathrm{Phase}}$. After quantization, the sensor signal frequency is 50 MHz, which gives a period *T* of 20 ns. One period equals 360${}^{{}^{\circ}}$, but from Figure 3, an effective phase range $\phi_{\mathrm{Sensor}}$ of 65${}^{{}^{\circ}}$ becomes apparent for the used sensor. The SNR for the phase evaluation is therefore calculated after
(3)$${\mathrm{SNR}}_{\mathrm{Jitter},\mathrm{Phase}}=10\;\cdot \;\log_{10}\left(\frac{\phi_{\mathrm{Sensor}}\;\cdot \;T}{360{}^{\circ }\;\cdot \;t_{\mathrm{Jitter},\mathrm{ADC}\;\mathrm{Aperture}}}\right)$$
and yields the result ${\mathrm{SNR}}_{\mathrm{Jitter},\mathrm{Phase}}=45.6$ dB. For an optimal sensor covering the full range of 360${}^{{}^{\circ}}$, the result is ${\mathrm{SNR}}_{\mathrm{Jitter},\mathrm{Phase}}=53.0$ dB, marking the best possible SNR for a phase evaluation based on the used ADC.

#### 3.1.3. Quantization Noise

An analog-to-digital conversion introduces quantization noise to the converted signal. The calculation of the root-mean-square (RMS) quantization noise voltage
(4)$$V_{\mathrm{QN}}=\frac{q}{\sqrt{12}}=\frac{V_{\mathrm{Ref}}}{\sqrt{12}\;\cdot \;2^{N}}$$
of an ideal ADC with the weight of a least significant bit *q*, which is the result of the ADC’s reference voltage $V_{\mathrm{Ref}}$ and its number of bits *N*, has been established before [28]. The general definition of an SNR is given by the signal power $P_{\mathrm{Signal}}$ and the noise power $P_{\mathrm{Noise}}$ according to
(5)$$\mathrm{SNR}=\frac{P_{\mathrm{Signal}}}{P_{\mathrm{Noise}}}=\frac{V_{\mathrm{Signal}}^{2}}{V_{\mathrm{Noise}}^{2}}=\frac{I_{\mathrm{Signal}}^{2}}{I_{\mathrm{Noise}}^{2}}\;.$$

Hence, the SNR due to quantization noise is calculated using Equations (Equation 4) and (Equation 5) according to
(6)$${\mathrm{SNR}}_{\mathrm{QN}}={\left(\frac{V_{\mathrm{Signal},\mathrm{eff}.}}{\frac{V_{\mathrm{Ref}}}{\sqrt{12}\;\cdot \;2^{N}}}\right)}^{2}\;.$$

For a sine wave as the signal, its peak signal voltage $V_{\mathrm{Signal},\mathrm{peak}.}=\sqrt{2}\;\cdot \;V_{\mathrm{Signal},\mathrm{eff}.}$ leads, when applied to Equation (Equation 6), to the SNR from quantization noise in decibel:(7)$${\mathrm{SNR}}_{\mathrm{QN}}=20\;\cdot \;\log_{10}\left(2^{N}\;\cdot \;\sqrt{6}\;\cdot \;\frac{V_{\mathrm{Signal},\mathrm{peak}}}{V_{\mathrm{Ref}}}\right)\;.$$

Two significant SNRs can be calculated, one for the lowest input signal level and one for the highest. For peak voltages of $V_{\mathrm{Signal},\mathrm{peak},\min}=41.50\;mV$ and $V_{\mathrm{Signal},\mathrm{peak},\max}=72.33\;mV$, an effective number of bits ENOB=9.8 [27] inserted for *N* and a reference voltage level $V_{\mathrm{Ref}}=1.2\;V$, the two SNRs result in ${\mathrm{SNR}}_{\mathrm{QN},\min}=37.6$ dB and ${\mathrm{SNR}}_{\mathrm{QN},\max}=42.4$ dB, corresponding to the quantization noise power of $P_{\mathrm{QN}}=-55.2\;\mathrm{dBm}$.

#### 3.1.4. Johnson–Nyquist Noise

The power of the Johnson–Nyquist noise $P_{\mathrm{JNN}}$ depends on the Boltzmann constant $k_{B}$, the measurement bandwidth BW and the temperature *T*. Following the Nyquist–Shannon sampling theorem and taking all bandwidth-restricting components of the setup into account, the measurement bandwidth is 150 MHz. Because the measurements were conducted at a temperature of 21 ${}^{{}^{\circ}}C$, calculating the noise power using
(8)$$P_{\mathrm{JNN}}=10\;\cdot \;\log_{10}\left(\frac{k_{B}\;\cdot \;T\;\cdot \;BW}{1\;\cdot \;10^{-3}W}\right)$$
leads to a value of −92.2 dBm.

#### 3.1.5. Relative Intensity Noise

The relative intensity noise (RIN) of the used DFB-LD is specified with better than −145 dB ${\mathrm{Hz}}^{-1}$ for the applied operating conditions. Considering the optical power of 10 dBm and the bandwidth of 150 MHz results in a RIN power of $P_{\mathrm{RIN}}=-53.2\;\mathrm{dBm}$.

#### 3.1.6. Shot Noise

Two sources of shot noise can be identified from the setup: the laser diode and the PD. The shot noise of the laser diode is part of its RIN, which has already been discussed. The shot noise of the PD depends on the photocurrent $I_{\mathrm{Photo}}$. The definition of the RMS shot noise current is
(9)$$\sqrt{\bar{i^{2}}}=\sqrt{2\;\cdot \;e\;\cdot \;I_{\mathrm{Photo}}\;\cdot \;BW}\;.$$

With Equation (Equation 5), the SNR due to shot noise is calculated by
(10)$${\mathrm{SNR}}_{\mathrm{SN}}={\left(\frac{I_{\mathrm{Signal},\mathrm{eff}.}}{\sqrt{\bar{i^{2}}}}\right)}^{2}\;.$$

The effective signal current, which is only one part of the total photocurrent, is calculated according to
(11)$$I_{\mathrm{Signal},\mathrm{eff}.}=\frac{V_{\mathrm{Signal},\mathrm{peak}}}{Z_{0}\;\cdot \;\sqrt{2G}}\;,$$
with characteristic impedance of $Z_{0}=50\Omega$ and gain of G=9 dB of the RF amplifier in the receiver block of the setup. Combining Equations (Equation 9)–(Equation 11) and transferring the finding to a decibel scale results in
(12)$${\mathrm{SNR}}_{\mathrm{SN}}=10\;\cdot \;\log_{10}\left(\frac{V_{\mathrm{Signal},\mathrm{peak}}^{2}}{4\;\cdot \;e\;\cdot \;{Z_{0}}^{2}\;\cdot \;G\;\cdot \;BW\;\cdot \;I_{\mathrm{Photo}}}\right).$$

Again, minimum and maximum values for the peak signal voltage have to be taken into account. The photocurrent equals the product of the PD’s sensitivity S=0.95 mA mW^−1^ and the power of the incident light $P_{\mathrm{Optical}}$:(13)$$I_{\mathrm{Photo}}=S\;\cdot \;P_{\mathrm{Optical}}.$$

It can be seen from the setup that $P_{\mathrm{Optical}}$ is the sum of the LD’s and EDFA’s output powers, which is less than the losses of the optical path.Altogether, $P_{\mathrm{Optical}}$ totals to not more than 7.8 dBm, which generates a photocurrent of 5.7 mA. With these values inserted, the ${\mathrm{SNR}}_{\mathrm{SN}}$ ranges from ${\mathrm{SNR}}_{\mathrm{SN},\min}=52.0$ dB to ${\mathrm{SNR}}_{\mathrm{SN},\max}=56.8$ dB, corresponding to the shot noise power $P_{\mathrm{SN}}=-69.6\;\mathrm{dBm}$.

#### 3.1.7. Further Influences

Some more characteristics influence the SNR. Additional amplitude noise can be introduced to the DFB-LD by the LDC. Measurements have shown an amplitude noise of the used custom LDC is not greater than the noise of typical commercial LDCs. Existing chirp of the DFB-LD has no effect on the measurement as the optical phase is not evaluated. The 6 dB splitter could have a temperature dependence, resulting in imbalanced amplitudes and phases for the branches. The evaluation scheme would tolerate such changes for the expected small gradients, which make the changes much slower than the measurement speed. Further on, the RF amplifier’s noise figure NF impacts the SNR. This affects only the SNR resulting from the Johnson–Nyquist noise, the relative intensity noise of the LD and the shot noise of the PD, which is following from the RF amplifier’s position in the signal path. Due to a high integration level of the receiver module, the noise figure of the used RF amplifier could not be specified as part of the present work. In addition, the ADC can theoretically introduce phase jitter between the two channels to the measurement. However, this is an unlikely mechanism as the datasheet indicates a fully parallel architecture. Finally, the sensor sensitivity has to be considered. Any relative shift in the FBG’s spectrum relative to the LD’s spectrum, independent from its cause (e.g., temperature changes, mechanical impacts), will deteriorate the measurement.

### 3.2. Discussion

Several noise sources have been identified and quantified for the observed setup up to this point. The total amplitude noise power $P_{\mathrm{AN}}$ resulting from the known contributions can be calculated according to
(14)$$P_{\mathrm{AN}}=\sqrt{P_{\mathrm{Jitter},\mathrm{ADC}}^{2}+P_{\mathrm{QN}}^{2}+{\mathrm{NF}}^{2}\;\cdot \;\left(P_{\mathrm{JNN}}^{2}+P_{\mathrm{RIN}}^{2}+P_{\mathrm{SN}}^{2}\right)}\;.$$

As the NF of the used RF amplifier is unknown, a typical value of 6 dB is assumed. Inserting this along with the previously calculated results into Equation (Equation 14) sums up to $P_{\mathrm{AN}}=-47.1\;\mathrm{dBm}$, resulting in a range of SNRs from ${\mathrm{SNR}}_{\mathrm{AN},\min}=29.5$ dB to ${\mathrm{SNR}}_{\mathrm{AN},\max}=34.3$ dB. It can be seen that the dominating noise contributions are the quantization noise and the RIN of the LD. Accordingly, some room for improvement is observable. An LD with a better RIN can be used. More advantages can be gained from matching the ADC input level better to the reference voltage, which improves the quantization noise, and from reducing the input frequency or using an ADC with a suitable analog bandwidth, leading to lower noise due to clock jitter. Finally, the shot noise can be reduced by roughly 67% solely by removing the sensor position monitoring because the EDFA contributes double the power to the total optical power compared to the LD.

As a consequence of the phase measurement being a relative evaluation, phase jitter of the modulating RF signal is inherently eliminated as long as the coherence length of the RF signal is not exceeded by the setup. The only degradation of the phase information is introduced by the ADC aperture jitter, leading to a signal-to-phase noise ratio of ${\mathrm{SNR}}_{\mathrm{Jitter},\mathrm{Phase}}=45.6$ dB. This can be improved by using an ADC with less internal aperture jitter or by optimizing the phase response characteristic of the FBG to cover a greater phase range, whereby an SNR improvement potential of more than 7 dB has been found for the latter.

It is clear from the previous considerations that the SNR of the amplitude evaluation greatly depends on the performance of many core components in the setup. An improvement in the SNR can be achieved only by costly changes to these components, especially to the LD and ADC but also to the RF amplifier stage. At the same time, using a reasonably selected sensor, the phase evaluation approach depends solely on the ADC performance. This involves not only a systematically better SNR but also offers the potential to select lower-quality components for the rest of the setup without degrading the SNR. For the investigated setup, the SNR of the phase evaluation approach is expected to be at least ${\mathrm{SNR}}_{\mathrm{AN},\max}-{\mathrm{SNR}}_{\mathrm{Jitter},\mathrm{Phase}}=11$ dB and at best more than ${\mathrm{SNR}}_{\mathrm{AN},\min}-{\mathrm{SNR}}_{\mathrm{Jitter},\mathrm{Phase}}=16$ dB better than for the amplitude evaluation approach. Albeit these values are specific to the used components, they highlight the finding that the phase evaluation is particularly advantageous over the conventional amplitude-based evaluation. Another benefit of the phase interrogation is the overall course of its characteristic curve due its broad linearity, which makes it generally less sensitive to relative shifts between sensor and laser sources.

## 4. Conclusions

This paper has explained the idea of basing the read-out of fiber sensors on the electrical evaluation of a microwave signal, which is introduced to modulate an optical source with a fixed wavelength, and distinguished this approach from the prior art of microwave photonics in sensing applications. An experimental setup and the measurement results have been presented. Two different characteristic curves of the same sensor can be interrogated with the approach: amplitude and phase. The setup has been analyzed for noise sources that degrade the microwave signal, and the consequences for either of the evaluation variants have been discussed.

The results indicate that strong advantages can be gained from using the phase-based approach because its only signal deterioration source is the ADC aperture jitter. An SNR gain of at least 11 dB has been achieved with the examined setup compared to the amplitude-based evaluation, which is considerably affected by more noise contributions. At the same time, it has been found that specific adjustments of the setup can improve the SNR for the amplitude-based measurement. These open the opportunity for further studies. Among the possible starting points are the quantization noise and the RIN of the optical source.

Furthermore, as has been addressed by the authors before [24], the implementation of this scheme in wireless sensing systems is of particular interest, as the already present microwave signal makes it predestined to adapt the concept for RoF measurement applications. After using the phase evaluation as the core of an RoF sensing system, even more advantages can be expected. This is because the phase of a wireless signal typically gets considerably less degraded during transmission than the amplitude of the same signal. Finally, the combined evaluation of phase and amplitude and accompanied potential information gain could be of interest for future studies. Further subjects of interest are the scheme’s sensitivity to the measurand, its response time and the maximum achievable sampling rate, among others.

## Figures and Tables

**Figure 1 sensors-23-03746-f001:**
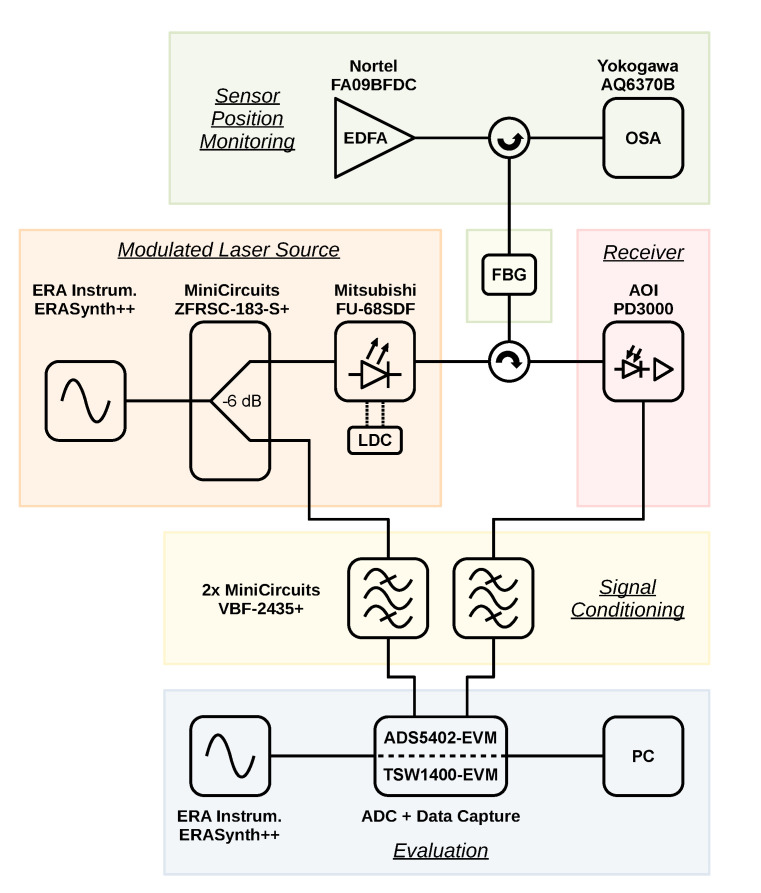
Block diagram of the experimental setup.

**Figure 2 sensors-23-03746-f002:**
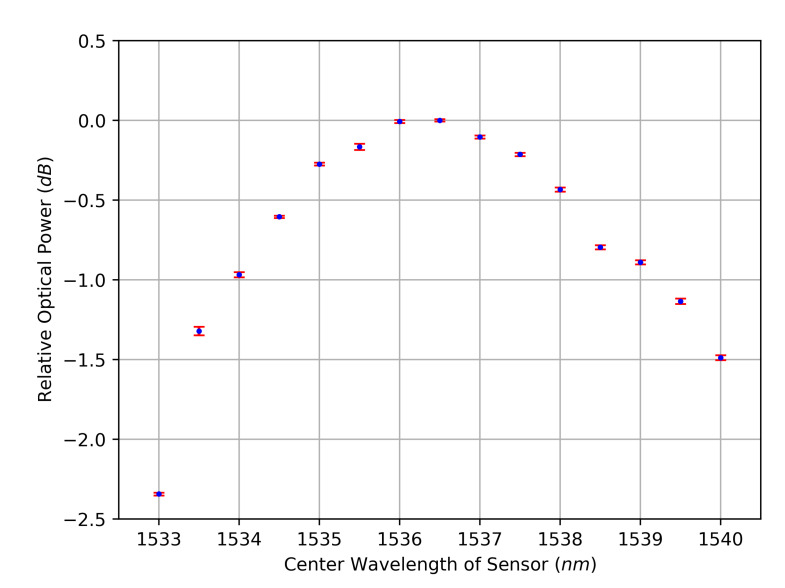
Measured dependency of the optical power from the center wavelength of the sensor. Mean (blue) and standard deviation (red) of 50 readings per sensor position.

**Figure 3 sensors-23-03746-f003:**
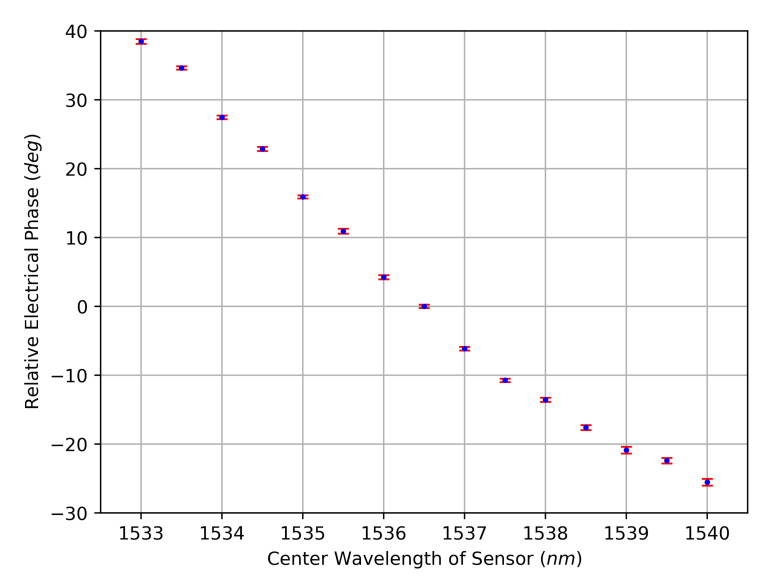
Measured dependency of the electrical phase from the center wavelength of the sensor. Mean (blue) and standard deviation (red) of 50 readings per sensor position.

## Data Availability

The data presented in this study are available on request from the corresponding author. The raw data are not publicly available due to limited relevance to the study.

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
