# Peer review of "On the Advantages of Microwave Photonic Interrogation of Fiber-Based Sensors: A Noise Analysis"

_sensors, 2023, doi:10.3390/s23073746_

Round 1

Reviewer 1 Report

Based on the application of microwave photonic method in optical fiber sensing, in order to distinguish it from the amplitude evaluation method, the authors presented a scheme for the interrogation of fiber sensors. However, neither has the measurement setup been analyzed nor have the amplitude and phase-based approaches been compared in detail before,this paper proposes a measurement device suitable for amplitude and phase evaluation of fiber Bragg gratings and studies the source of signal degradation, showing the advantages of signal-to-noise ratio of phase response evaluation. This paper has certain application value.  The following questions suggest further modifications.

(1) In the abstract part,the author twice stated that "a scheme for the interrogation of fiber sensors which is based on a fiber Bragg grating’s phase response for the electrical signal." and "this paper proposes a measurement setup suitable for amplitude and phase-based evaluation of fiber Bragg gratings" It is recommended to further refine it to make it clearer;

(2) The abstract indicates that "this paper mainly proposes a fiber optic sensor detection scheme for the phase response of fiber grating to electrical signals", but the keyword "phase response" does not appear. it is recommended to supplement and make modifications;

(3) In section 2.1, the formulation of basic concepts is classical, it is recommended adding quotations;

(4)  In Figure 1 of Section 2.2, the author shows the block diagram of the experimental device, however, there are individual components that are not indicated. The author make some additions appropriately;

(5) In the result analysis of section 2.4, the depth of analysis of the measurement results of amplitude features and phase features is insufficient, and the expression "Judging by the available resolution, the phase characteristic seems to be an injective function" is vague, it is recommended adding modifications;

(6) In section 3.1, there is a phenomenon that the punctuation marks after the formula are not uniform, and it is recommended that the author makes modifications to standardize the writing;

(7) In section 3.1.2, "φSensor" is not indicated in the previous text, and it is recommended making some commentsï¼›

(8) In the formula (4) of section 3.1.3, there is no explanation for the symbol that appears for the first time in the article, and the results of a large number of data are based on previous research, which is not easy to understand in the process of derivation. It is recommended to simplify the processing;

(9) In section 3.1.4,The "Boltzmann constant kB" appears for the first time in an article, but does not make a comment

(10) Section 3.1.7 states that "ADCs can theoretically introduce phase jitter between two channels for measurement, but an unlikely mechanism", which is not explained by exact data in this article;

(11) The conclusion of this paper mentions that "compared with amplitude-based evaluation, the test setup obtains at least 11dB of signal-to-noise ratio gain", but there is no obvious data interpretation or deduction during the data analysis process. It is recommended to supplement.

In summary, it is recommended that this article needs to be revised and reconsidered.

Reviewer 2 Report

In this manuscript, the authors proposed a measurement setup suitable for amplitude and phase-based evaluation of fiber Bragg gratings. The research work has certain theoretical and practical value. But as the author said, there are some articles in this research field. My comments are as follows.

1.The innovation of this manuscript is not obvious. There are several works proposed [21]-[26]. What are the advantages of the proposed scheme in this manuscript?

2. Compared with the traditional amplitude method, what are the applicable conditions? What do they cost?

3. In Figure 2 and Figure 3, how many times have the experimental data been measured?

4. There are too many formulas in the manuscript.In this manuscript, the authors proposed a measurement setup suitable for amplitude and phase-based evaluation of fiber Bragg gratings. The research work has certain theoretical and practical value. But as the author said, there are some articles in this research field. My comments are as follows.

1.The innovation of this manuscript is not obvious. There are several works proposed [21]-[26]. What are the advantages of the proposed scheme in this manuscript?

2. Compared with the traditional amplitude method, what are the applicable conditions? What do they cost?

3. In Figure 2 and Figure 3, how many times have the experimental data been measured?

4. There are too many formulas in the manuscript.

Reviewer 3 Report

Authors successfully proposed a measurement setup suitable for amplitude  and phase-based evaluation of fiber Bragg gratings and investigates for sources of signal degradation, an aspect that has not been considered before. Several comments:

1-The title "On the Advantages of Microwave Photonic Interrogation of Fiber-based Sensors" is not particularly describe the overall content of the paper. Suggest to revise the title.

2-The finding is more on noise analysis. Suggest to revise the title containing noise analysis.

3-Despite of noise characteristic, many other sensing performance parameters need to be considered such as sensitivity, response time etc. Suggest to further study in future.

Round 2

Reviewer 1 Report

All are revised based on the comments.

Author Response

Thank you for the feedback.

Reviewer 2 Report

In this new version, the manuscript was improved, and the authors addressed the comments. But there are some points should be emphasized.

1.there are many formulas, and it is recommended to merge them. For example, formulas (4) - (6) are general formulas, and it is recommended to delete them.

2.The conclusion needs improvement.

3.Authors should carefully check the manuscript before submission.
